# Student Stress and the Effects of Relaxation: A Study Conducted at the University of Lille in Northern France

Dan Gondo, Denis Bernardeau-Moreau *  and Philippe Campillo

Multidisciplinary Research Unit for Sport, Health, Society (URePSSS), University of Lille, ULR 7369, 59000 Lille, France; gondo.dan@univ-lille.fr (D.G.)
* Correspondence: denis.bernardeau-moreau@univ-lille.fr

**Abstract:** Although all sections of society experience periods of stress of varying intensity, there is one population that appears to be particularly vulnerable to stress and its harmful effects: students. Many studies attest to the high level of vulnerability experienced by this young and fragile population, exposed to situations that generate tension, doubt, and stress. Studies estimate that one in three young people suffer from somatic and emotional disorders and have difficulty managing their stress. What can be done about this situation? Based on a representative sample of students enrolled at the University of Lille (Northern France), our study aims to more accurately identify stress levels and factors among the student population. To do this, we conducted three surveys: a qualitative survey consisting of interviews (N = 165), a quantitative survey based on a questionnaire (N = 1049) and an immersive survey where students were invited to take part in an initial relaxation session (N = 22). For the interviews, we used the PSS (Perceived Stress Scale) method of Cohen et al. For the questionnaires, we used the self-administered method (questionnaires sent out and returned by email) with the online application "Survio". We have thus tried to better assess the impact of sport and relaxation on student stress and determine the extent to which these activities can significantly reduce stress and its harmful effects on the health of young students. Our results show that the main stress factors are exams and uncertainty about the future. The main symptoms of stress are tachycardia and stomach ache. To manage their stress, students prefer to play sports and to go out with friends. Finally, we show a significant correlation between the practice of relaxation and the reduction in stress levels.

**Keywords:** student stress; exams; benefits of sport; effect of relaxation



## 1. Introduction

Stress is defined as a psychological process developed by the body in response to a situation or event that is particularly stressful or perceived as such. If prolonged, it can lead to long-lasting psychological and physical exhaustion that may negatively impact the individual's social life (Buitekant 2019). Although all sections of society experience periods of stress of varying intensity (9 out of 10 French citizens suffer from stress and claim to have experienced stress at some point in their life, according to a study conducted by Laboratoire Lescuyer in 2022),[1] there is one population that appears to be particularly vulnerable to stress and its harmful effects: students. According to the website of the French Ministry of Higher Education and Research (MESR), the student population in 2020–2021 represented over 1.65 million students enrolled in French universities. This population is growing at a steady rate. The year 2021 saw a 0.9% increase in student enrolment compared to the previous academic year and a 1.7% increase in the number of students enrolled in their first year of higher education.[2] According to the Ministry's website, the student population has grown by nearly 220,000 students per year, representing an average growth of 2.2% per year over the last five years in France.[3] This increase in the student population places France in 21st place among OECD countries for its proportion of higher education graduates at

full working age, i.e., between 25 and 64 years old (SIES 2022).[4] Although the student population is not diminishing, the economic, social, and political context in which it is growing does not appear to be conducive to its healthy development. Many studies attest to the high level of vulnerability experienced by this young and fragile population, exposed to situations that generate tension, doubt, and stress.

Several studies conducted on the subject of student stress show high levels of stress, especially among younger students. Indeed, at least one-third of the population studied is concerned by stress. The studies also point out that the pandemic and the conditions of isolation have greatly worsened the already worrying situation of students. According to Beck et al. (2006, p. 4), these depressive symptoms are experienced on a daily basis by 22% of students (compared to 11% in the general population). In his study on mental health among students, Fisher (1994) estimates that one in three young people who have left the family home suffer from somatic disorders (various types of pain, digestive disorders, etc.) and emotional disorders (anxietyy, depression, etc.). The report of a field visit carried out at the request of the Île-de-France Regional Health Agency (ARS) by Dr. Jean-Christophe Maccotta and Prof. Maurice Corcos[5] also concludes that one-third of French students show signs of psychological distress. Other studies confirm this proportion of the student population is affected by stress. A survey by the regional student mutual insurance companies (2007)[6] shows that in France, one-third of students (36.2%) admit to struggling to manage their stress. The same percentage (31.3%) describes having experienced, for a period of more than two weeks, moments of sadness, depression, and despair (OVE 2018).[7] It is worth noting that these signs are often the first criteria for diagnosing a major or confirmed depressive episode. According to the 2015 MESR study,[8] 54% of students frequently experience symptoms of stress. Another study (Romo et al. 2019) shows that among the most common issues encountered in the young population, mood disorders, depression, anxiety, stress, and psychological distress are the most frequently cited. According to BVA 2020,[9] two out of three young people (64%) say they are stressed because of their studies (it should be noted that these are mainly high school and university students). Another point to note is that stress seems to be poorly taken into account by the students. Indeed, according to the above-mentioned ARS report, only 30% of students experiencing poor mental health consult a general practitioner, psychiatrist, or psychologist. The 70% who do not seek help are either unaware of their suffering or are unaware of the care facilities available. According to some international studies (Nerdrum et al. 2006; Tyrrell and Smith 1996), student stress rates in France are similar to those of other European countries. Furthermore, studies also show that the most vulnerable or most affected students are the youngest students, especially those studying undergraduate programmes (OVE 2018). According to Mazé and Verlhiac (2013), university entry and the high repetition rates during the three years of the undergraduate programme—(according to Goulard (2007),[10] the rate of students entering their third year after two years of study across all programmes, is only 37%)—explain the vulnerability facing younger students. According to Saleh et al. (2017), the presence of these psychological difficulties is associated with poorer academic performance and lower professional performance in the future.

As student health was already a concern, the global pandemic generated by the mass, brutal spread of COVID-19 has only made an already tense and difficult situation even worse. Initial studies show that the pandemic has had a significant impact on the lives of the population, particularly young people. According to a recent survey carried out for L'Étudiant,[11] the first lockdown in France had a major effect on students, causing the majority of them to suffer increasingly poor mental health. The Fages-Ipsos study (2020)[12] concludes that 76% of young people and 83% of students say they have been affected psychologically, emotionally, or physically by the pandemic. The study argues that this level of distress is particularly alarming as more than a quarter of these young people (27%) and more than a third of students (31%) say they have experienced suicidal thoughts since the beginning of the crisis (a percentage that has risen sharply in 9 months). A study published in 2021 (Macali et al. 2021) shows similar results. During the lockdown

period, students were more affected by mental health issues than non-students of the same age (37% and 20%, respectively). According to the study conducted by the French National Centre for Resources and Resilience (CNRR),[13] 43% of students surveyed were suffering from mental health issues after the first lockdown. Symptoms were various: isolation, malaise, increasingly severe social insecurity, digital divide, inequalities in living conditions during the lockdown period, etc. All of these issues reinforce and exacerbate the inequalities that already existed prior to the pandemic. According to the French National Institute of Statistics and Economic Studies (INSEE),[14] 20% of students in France used to be reported as living below the poverty line, but a report by the Federation of General Student Associations (FAGE), published in 2020,[15] emphasises that the precarious situation of students has worsened due to the COVID-19 pandemic. The closure of university canteens offering affordable meals also led to the elimination of many student jobs. One in three students admits to having forgone medical care due to a lack of financial means (OVE survey 2018 quoted above). In addition to all these difficult situations, the population faced abrupt changes to social life (obligation to respect health restrictions, the introduction of e-learning classes). According to the FAGE (Ipsos 2020 survey), the various sources of stress highlighted by students were the reduction in social interaction, uncertainty regarding their future, the methods involved in remote learning, and the high level of insecurity due to the elimination of many student jobs. According to a 2021 MILDECA survey,[16] the pandemic has considerably encouraged risky behaviour in the general population (alcohol consumption, drunk driving, prolonged sedentary lifestyle, etc.), particularly among young people.

All these above-mentioned studies clearly show that students are an at-risk population that is particularly exposed to stress in its various forms. These forms of stress are detrimental to their well-being and lead to potentially serious psychological and cardiovascular diseases. To understand the stress and its consequences, Lassarre et al. (2002) point out that we must take into account which periods are particularly conducive to stress. Some surveys (Manolova et al. 2012; Doron et al. 2012; Saleh 2017) show that stress levels are particularly high at the beginning of the year, during exam periods, and while waiting for grades to be announced. Aware of this increasingly serious issue, universities have set up medical check-up procedures (access to free healthcare on university campuses). They have also organised social activities, such as sports, going out with friends, music, and relaxation techniques. However, in light of what is demonstrated by the above-mentioned surveys, are these measures enough? Are they useful in preventing the most serious stressful situations?

Our intention in this article is to examine the phenomenon of student stress and the possible ways of alleviating it, particularly through relaxation practices. Numerous studies establish a strong correlation between stress and the mitigating effects of sport and relaxation (Cruz et al. 2013; Boujut and Décamps 2012; Gómez-Gallego et al. 2020; Vidic et al. 2017). In the intention of achieving harmony between body and mind, these practices can certainly be used as preventive and/or curative therapy, but are they sufficiently effective to alleviate students' stress levels in the long term? Are students aware of these activities, and do they take part in them often enough? Based on a representative sample of students enrolled at the University of Lille (Northern France), our study aims to more accurately identify stress levels and factors among the student population. First, we want to assess stress among students in Hauts-de-France and see if our survey confirms previous studies in this area. What are the stress factors? How is it expressed? What are its symptoms? In the second phase, our aim is to better assess the impact of sports and especially relaxation on student stress and determine the extent to which these activities can significantly reduce stress and its harmful effects on the health of young students.

## 2. Materials and Method

In order to fully understand student stress, we felt it was necessary, in the context of this study, to first examine the factors, symptoms and consequences of stress, and secondly the tools for managing it more effectively, with a particular focus on relaxation methods.

To do this, we conducted three surveys: a qualitative survey consisting of interviews (N = 165), a quantitative survey based on a questionnaire (N = 1049) and an immersive survey where students were invited to take part in an initial relaxation session (N = 22). The choice of students from Hauts-de-France in the north of France is not a coincidence. The authors of this study are teachers in different departments of this university. Their professional position facilitated the contacts for the interviews and the immersion session. The University of Lille is representative of universities in France, particularly in terms of the number and socio-economic profile of its student population.

First, we conducted a series of interviews using the method developed by Cohen et al. (1983). The PSS (Perceived Stress Scale) is a scale used to measure perceived stress based on the transactional approach, and engaging the psycho-cognitive mechanisms. The interviews, which lasted 20 min each (see the standard model in Appendix A), took place between March and April 2022, involving students enrolled in their first year of various degree programmes (sports science, medicine, business school, engineering school, and law-Table 1). As our intention was to measure stress among students from different educational backgrounds, we composed a sample of N = 165 students divided into two groups: a control group and an experimental group. The first group is made up of students in their first year of a university degree in sports science (in the degree programme known in France as "Staps"). This reference group (or control group) consists of 82 respondents, 40% of whom are girls and 60% boys. The second group is composed of 83 first-year students enrolled in several degree programmes, distributed as follows: 20 medical students, of whom 60% are boys and 40% girls; 21 law students, of whom 43% are boys and 57% girls; 22 business students, of whom 54% are boys and 46% girls; and 20 engineering students, of whom 55% are boys and 45% girls. These specific degree programmes were not selected at random. They were selected based on their entry and examination procedures. Some are very selective, others much less so, and the level of competition between students differs according to the degree programme. It should be noted that the age of the students selected for the interview ranged from 17 to 25 years. It should also be noted that a dictaphone was used to conduct the qualitative study. For the audio transcriptions, we used the software TRINT, which automatically transforms audio into text.

**Table 1.** Characteristics of the qualitative study sample group (N = 165 interviews).

| Group by Degree Programme | Number of Respondents | % Compared to Total Sample Group | Boy/Girl Ratio |
|---|---|---|---|
| Sport sciences | 82 | 49% | 50/32 |
| Medecine | 20 | 12% | 12/8 |
| Law | 21 | 13% | 9/12 |
| Engineering School | 20 | 12% | 11/9 |
| Business School | 22 | 14% | 19/12 |
| Total | 165 | 100% | 92/73 |

Second, we carried out a quantitative study. The questionnaire survey was carried out during the 2020–2021 academic year and involved 1049 respondents, all enrolled in the first year of a sports science ("Staps") degree. We chose this programme because it is the course with the most students who actively practise or have practised relaxation. In this respect, this may allow for a useful and relevant comparison between practitioners and non-practitioners. The students in this course are between 17 and 25 years old and are 60% male and 40% female. Our study used the same questions as the qualitative analysis but used the self-administered method (questionnaires sent out and returned by email). The online application called "Survio" was used to create the questionnaires that could be accessed directly from a phone or computer. Responses were collected via this application. An Excel table and statistical graphs were generated, allowing for several possible cross-tabulations between the different variables. The "Tropes" and "Iramuteq" software packages were used to process certain responses and identify trends. To measure

the reliability of the results obtained from the comparative analysis of the values, we used the Student's *t*-test.

In the third phase of the study, we conducted a relaxation session with first-year students enrolled in an engineering school in Lille. Two reasons explain the choice of this field of study. First, the engineering school showed great interest in organising relaxation sessions for its students (this project is very innovative for this institution). Second, it was important for us to work with students who had never practised relaxation in order to better measure the effects of this activity on the feeling of stress. The engineering school had this double advantage. The aim was to assess the students' stress levels at the start of the academic year (T0) and after a relaxation session (T1)—it is worth noting that our intention is to continue the relaxation sessions on a quarterly basis for several years with the same group in order to measure the long-term effects of relaxation on stress. This study took place during the second term of the academic year and involved 22 students (40% girls and 60% boys). A stress identification and assessment questionnaire based on Cohen et al. (1983) was provided to the students beforehand. Students then participated in a 75 min relaxation session based on the body scan method. The same questionnaire was given to the students after the session to get their feedback. The online software Survio was used to collect the responses directly and analyse the results.

## 3. Results

### 3.1. Stress Factors: Unequal Situations in the Face of Pressure

Some previous studies have examined stress factors in schools (Cicchelli 2001; Fisher 1994; Lassarre et al. 2002; Lassarre and Paty 2003). Based on these studies and their methodology, we interviewed an initial cohort of students (n = 20) using open-ended questions to establish a list of primary stress factors. This first approach resulted in the identification of only seven factors considered to be significantly linked to stress, i.e., whose representativeness rate exceeded the 5% threshold. These factors are exams, uncertainty about the future, change of institution, administrative organisation, personal organisation, workload, and competition between students. Then, we conducted interviews with our entire sample group (N = 165). The results show that stress is experienced differently by students according to the field of study (Table 2). Indeed, while overall, students in sports science and business degree programmes are not very stressed (only 14.36% and 13.2%, respectively say that they feel stressed very often), those in the medicine, law and engineering courses show higher, even much higher scores (31.50%, 24.25%, and 16.69%, respectively), with the medical students being the most stressed.

**Table 2.** Comparison of rates of students reporting very frequent stress by degree programme.

| Group by Degree Programme | Number of Respondents | Frequencies (SF) | Averages (SFF) | Percentages (SFF) |
|---|---|---|---|---|
| Sport sciences | 82 | 187 | 2.28 | 14.36 |
| Medecine | 20 | 100 | 5 | 31.50 |
| Law | 21 | 81 | 3.85 | 24.25 |
| Engineering School | 20 | 53 | 2.65 | 16.69 |
| Business School | 22 | 46 | 2.09 | 13.2 |
| Total | 165 | 467 | 15.87 | 100 |

SF: Stress factors/SFF: Stress factor frequency. Table legend: For the sports sciences students, the values "stress factors" appear 187 times in the interviews and 2.28 times on average per value for an overall percentage of 14.36% of stressed students.

After noting that the distributions of the variables did not follow a normal distribution (Shapiro–Wilk test), the application of the non-parametric Kruskal–Wallis ANOVA test by rank reveals strong significant differences at $p < 0.001$ between the disciplinary groups. Multiple comparisons ($p$ values and z' values) of the sum of stress factors characterise and confirm ($p < 0.001$) significant differences mainly between the Medicine and Law groups with the other groups, STAPS, engineering, and business schools.

The analysis of the responses (Table 3) shows, moreover, that exams remain the main source of stress for all students (87.87%). This was followed by uncertainty about the future (40.6%) and changing institutions (39.39%—note that this was mainly described as the jump from high school to university). The administrative organisation is also a problem for students who complain about procedural constraints and difficulties in obtaining information about their degree programme (26.06%). The workload is also a major generator of stress for students (21.21%). However, this stress factor is most prevalent among medical and law students (75% and 61.48%, respectively). Although students studying sports science appear to be less stressed, this difference can be explained by their lighter workload. This factor is cited 32 times by students in other fields of study but only three times by students in this field. Personal organisation (10.9% of respondents), on the other hand, appears to be more of a problem for engineering students while not posing any particular issue among business students. It should also be noted that only medical students recognise that competition between students is a source of significant levels of stress (this is the case for 70% of medical students, but it is true that selection is a key aspect of this degree programme).

**Table 3.** Comparison of student stress factors by degree programme.

| Groups | Respondents | Sexe | Age | 1. Exams | 2. Uncertainty about the Future | 3. Changing Institutions | 4. Administrative Institutions | 5. Workload | 6. Personal Organisation | 7. Competition between Students |
|---|---|---|---|---|---|---|---|---|---|---|
| Sport Sciences | 82/49% | G:50 F:32 | 17–23 | 74 90.24% | 22 26.82% | 29 35.36% | 21 25.60% | 3 3.65% | 7 8.53% | 0 0.00% |
| Medecine | 20/12% | G:12 F:08 | 17–19 | 18 81.81% | 16 80% | 12 60% | 8 40% | 15 75% | 2 10% | 14 70% |
| Law | 21/12,72% | G:09 F:12 | 17–25 | 17 80.95% | 18 85.71% | 11 52.38% | 5 23.80% | 13 61.48% | 3 14.28% | 0 0.00% |
| Engineering School | 20/12% | G:11 F:09 | 17–23 | 16 80% | 6 30% | 10 50% | 5 25% | 2 10% | 6 30% | 0 0.00% |
| Business School | 22/13.33% | G:10 F:12 | 18–19 | 20 90.90% | 5 22.72% | 3 13.63% | 4 18.18% | 2 9.09% | 0 0,00% | 0 0.00% |
| Total | 165/100% | G:92 F:73 | 17–25 | 145 87.87% | 67 40.60% | 65 39.39% | 43 26.06% | 35 21.21% | 18 10.90% | 14 8.48% |

Table legend: For sports sciences students, exams are the first stress factor (for 90.24%), followed by uncertainty about the future (26.82%).

### 3.2. Stress Symptoms: When the Body Speaks Out

Our study also shows that students experience the same symptoms of stress regardless of their field of study (Table 4). The main symptoms reported by students are tachycardia (accelerated heart rate) 55.15%, stomach ache (33.93%), and insomnia (26.66%). This is followed by overthinking (24.24%), trembling hands and legs (19.39%), and feeling hot (16.96%). Other less common factors are also highlighted, such as a frequent need to go to the toilet, lack of concentration, loss of capacities, and a feeling of having sweaty hands.

**Table 4.** Comparison of student stress symptoms by degree programme.

| Groups | Respondents | Sexe | Age | 1. Tachycardia | 2. Stomach Ache | 3. Insomnia | 4. Overthinking | 5. Trembling Hands and Legs | 6. Feeling Hot | 7. Other |
|---|---|---|---|---|---|---|---|---|---|---|
| Sport Sciences | 82/49% | G:50 F:32 | 17–23 | 50 60.97% | 20 24.39% | 19 23.17% | 23 28.04% | 12 14.63% | 15 18.29% | 10 12.19% |
| Medecine | 20/12% | G:12 F:08 | 17–19 | 11 55% | 8 40% | 4 20% | 5 25% | 5 25% | 1 5% | 1 5% |
| Law | 21/12.72% | G:09 F:12 | 17–25 | 11 52,38% | 10 46.61% | 6 28.57% | 6 28.57% | 5 23.80% | 2 9.52% | 2 9.52% |
| Engineering School | 20/12% | G:11 F:09 | 17–23 | 11 55% | 9 45% | 6 30% | 4 20% | 4 20% | 5 25% | 1 5% |
| Business School | 22/13.33% | G:10 F:12 | 18–19 | 8 36.36% | 9 40.90% | 9 40.90% | 2 9.09% | 6 27.27% | 5 22.72% | 5 22.72% |
| Total | 165/100% | G:92 F:73 | 17–25 | 91 55.15% | 56 33.93% | 44 26.66% | 40 24.24% | 32 19.39% | 28 16.96% | 19 11.51% |

Table legend: For sports sciences students, the first symptom of stress is tachycardia (60.97%), followed by stomach ache (24.39%).

Thus, if stress is a frequent and widespread phenomenon among the cohorts of students taking part in the study (but, as we have seen, to varying degrees depending on the degree programme), how do they manage it and what methods do they use?

### 3.3. Stress Management: A Variety of More or Less Proven Strategies

When dealing with stress, it is interesting to find that students use a variety of strategies to manage and reduce it as much as possible (Table 5). When asked what means they use to combat stress, most students (70.9%) say they play sports. This percentage is highest in the sports science degree programme (95.12%) and very high in the other programmes (between 45 and 50%). A third of students, all fields of study included, prefer to go out with friends (33.93%) or listen to music (alone or at a concert, 32.12%). Relaxation, breathing, and sophrology techniques only come fourth, with 29.09% of respondents describing using these methods. A quarter of respondents (23.03%) were able to take a step back, distance themselves from their problems and put things into perspective. Finally, students describe communicating with relatives and the need to confide in them (21.81%) and taking medication (9.69%). Again, the latter appears to be a particular phenomenon among medical students, 65% of whom (!) report taking medication to combat stress.

**Table 5.** Comparison of student management methods by degree programme.

| Groups | Respondents | Gender | Age | 1. Practice Sports | 2. Go Out with Friends | 3. Listen to Music (Alone or at a Concert) | 4. Relaxation, Breathing and Sophrology Techniques | 5. Take a Step Back | 6. Communicate with Friends, Confidants | 7. Take Medication |
|---|---|---|---|---|---|---|---|---|---|---|
| Sports Science | 82 49% | M:50 W:32 | (17–23) | 78 95.12% | 23 28.04% | 23 28.04% | 28 34.14% | 14 17.07% | 9 10.97% | 1 1.21% |
| Medicine | 20 12% | M:12 W:08 | (17–19) | 8 40% | 8 40% | 9 45% | 5 25% | 4 20% | 9 45% | 13 65% |
| Law | 21 12.72% | M:09 W:12 | (17–25) | 12 57.14% | 10 47.61% | 11 52.38% | 3 14.28% | 8 38.09% | 5 23.80% | 0 0.00% |
| Engineering School | 20 12% | M:11 W:9 | (17–23) | 9 45% | 9 45% | 5 25% | 6 30% | 8 40% | 7 35% | 1 5% |
| Business School | 22 13.33% | M:10 W:12 | (18–19) | 10 45.45% | 6 27.27% | 5 22.72% | 6 27.27% | 4 18.18% | 6 27.27% | 1 4.54% |
| Total | 165 100% | M:92 W:73 | (17–25) | 117 70.90% | 56 33.93% | 53 32.12% | 48 29.09% | 38 23.03% | 36 21.81% | 16 9.69% |

Table legend: For sports sciences students, sport is the first stress management factor (95.12%), followed by going out with friends (28.04%).

As far as relaxation is concerned, we observed that in degree programmes with the lowest stress levels (business 13.3%, sports science 14.36%, and engineering 16.69%), the rates of relaxation practice seem to be the highest (27.27%, 34.14%, and 30%, respectively). Could relaxation activities be an effective way of reducing the stress levels experienced by students? Cross-sorting the "stress level" and "relaxation practice" data seems to reveal an interesting correlation.

### 3.4. Relaxation: A Source of Calm and an Effective Way to Combat Stress

The objective of the quantitative analysis by questionnaire was to verify the possible correlation between the practice of relaxation and stress levels among of students. For this, we interviewed N = 1049 students in their first year of a sports science ("Staps") degree programme. As relaxation techniques incorporated into the degree programme were an option to be chosen from among other possible options, this arrangement maximised the return rates and the comparative study (Table 6). As can be expected, sports science students practise a lot of sports activities (football for 22% of the respondents, basketball for 14%, weight training for 11%, and athletics for 9%). Most of these students (74%) have a proficient level in their chosen sport and practise it as part of a club. Eighty-one per cent of students report having practised the sport for more than three years. When asked the question, "in the past six months, how often have you felt stressed: never or very frequently?", 41% of respondents answered that they never felt stressed, 42% often,

and 17% admitted to having felt stressed very often. It should be noted that these rates (similar to those obtained in the qualitative survey, which are quite similar) remain well below the national average. When it comes to relaxation techniques, although more than half of students had never practised these methods (58%, of which one-third are girls and two-thirds are boys), does our study reveal relaxation to have a beneficial effect on reducing stress among students? To answer this question, we performed a Kuskal-Wallis ANOVA test by rank associated with a Cki-Square test to compare stress levels between practitioners and non-practitioners of relaxation. Our analysis confirms that there is a significant difference between practitioners and non-practitioners of relaxation in terms of their stress levels ($p < 0.01$). Indeed, among the students who report never having felt stressed, 53.1% practise relaxation (of which 43% are girls). However, of the students who say they are very frequently stressed, only 26.9% practise relaxation (of which 15% are girls). As can be seen, the group reporting never having felt stressed had the highest proportion of students practising relaxation techniques. On the other hand, the majority of students who report very frequently stressed do not practise relaxation.

**Table 6.** Kruskal–Wallis ANOVA by Ranks and Chi-Square tests between the "stress" factor and the "practises relaxation" factor.

| Stress Level | Students Who Do Not Practice Relaxation (%) | Students Who Practice Relaxation (%) | Correlation Coefficient |
|---|---|---|---|
| Never stressed | 46.9% | 53.1% | r = 0.579 * |
| Very frequently stressed | 73.1% | 26.9% | r = 0.579 * |

Table legend: Among the students who never or very rarely stressed, we find a majority of relaxation practitioners (52%). Among students who are often and very frequently stressed, the majority are non-practitioners of relaxation (67%). * $p < 0.05$.

## 4. Discussion

Ultimately, our results seem to highlight a positive correlation between regular relaxation practice and lower stress levels. This correlation requires further discussion, as outlined below.

We surveyed the students who described practising relaxation to find out why they practise it and what words they associate with it. According to our respondents, relaxation techniques bring about feelings of satisfaction, positive emotions, calmness, and relaxation (see Figure 1). They promote self-reflection and optimise concentration, as shown by the analysis conducted using Tropes (semantic analysis software), which allows us to identify units of meaning (satisfaction, calm, concentration, relaxation, and self-reflection).

As the following quotes show, relaxation is seen by its practitioners as a beneficial activity and a useful way to manage stress:

1. it's relaxing, that's all, it's nice and relaxing. 2. it's good for your body, breathing techniques, body wellness, relaxation. 3. a quiet moment to clear your head. 4. good for your health, it relaxes you, it makes you more focused. 5. It helps you unwind, useful during stressful or tiring periods, 6. a quiet moment to clear your head, good for your health. 7. helps to relax, good in times of stress or fatigue. 8. it helps in your day-to-day life, good for the body, allows your body to rest. 9. in your head, can be used for recovery, it's soothing. 10. it helps to boost your well-being.

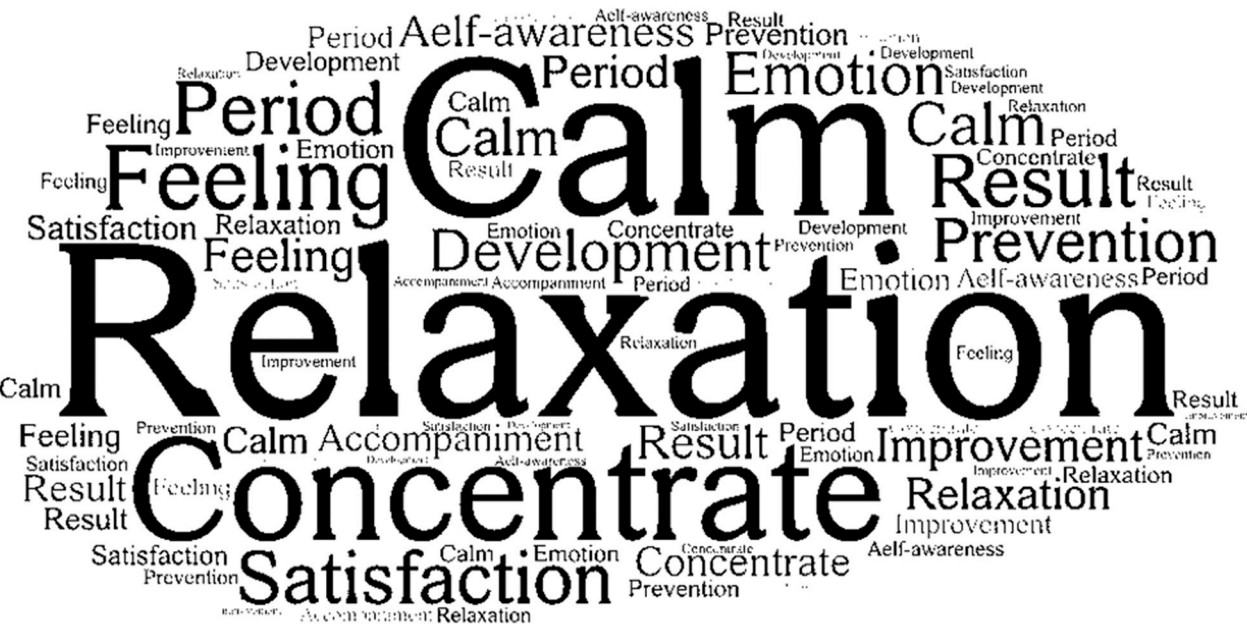

**Figure 1.** Spectrums of meaning: What do you gain from practising relaxation?

### 5. The Immediate Effects of a Relaxation Session on Feelings of Stress

Some higher education institutions (Nice, HEC Business School) have been offering relaxation programmes for several years to combat exam-related stress and blank page syndrome. These initiatives are based on various projects aimed at measuring the impact of relaxation techniques or self-care sessions on stress level among students. Deckro et al. (2002) organised relaxation and behavioural and cognitive skills classes for students (these sessions lasted 90 min and took place over a period of one and a half months, with one session per week). The results showed a significant reduction in psychological distress, anxiety, and perceived stress in these students. Bughi et al. (2006), by following a brief behavioural intervention programme or BBIP, which uses deep diaphragmatic breathing, relaxation, and active meditation as stress management techniques, show that the prevalence of stress in medical students decreases by almost 50% after a course of sessions lasting several weeks. A study by McGrady et al. (2012) measured the effects of a wellness programme on levels of anxiety and depression in a cohort of 450 medical students. A significant and sustained decrease in depression levels was observed among the students. Other studies of a similar nature (Jain et al. 2007; Yusoff et al. 2011; Ratanasiripong et al. 2012) all show identical results. In his 2015 feasibility study of the stress management programme, Saleh (2017) highlights the value of using an online intervention programme to improve self-esteem and significantly reduce psychological distress.

In our study, we carried out an initial relaxation session to measure student stress level at T0 (i.e., before the session) and T1 (i.e., after the session). This situation test was only intended to illustrate the correlation coefficient highlighted in our article between stress level and the effects of relaxation. The implementation of a long-term survey protocol has been planned in order to measure the perceived beneficial and constant effects of relaxation on the reduction in stress levels in the long term. The session was designed to incorporate the Jacobson method and body scan sophrology. The group was composed of 22 students (60% boys and 40% girls) enrolled in their first year of engineering school. The first questionnaire was given to them two weeks before the session (T0). This questionnaire was based on the Cohen et al. perceived stress scale or PSS (1983). Responses were collected online via the Survio application. The session took place on 02/12/2022. It lasted one hour and was followed by a 15 min discussion. At the end of the relaxation session, students were given a second questionnaire (T1).



At T0, three groups stand out when referring to the Cohen et al. perceived stress scale (Table 7). The first group (27%) includes students with a PSS score of over 27. According to this scale, this means that life for this population poses a constant threat. Most situations instil them with a sense of powerlessness that generates discomfort and even pathologies. The second group (36%) had a PSS score of 21 to 26. These are students who have a moderate level of perceived stress and are generally able to cope with their stress. Finally, the third group (36%) had a PSS score of less than 21. These are students with very low perceived stress and who can easily adapt to the situations they are faced with. At the end of the first relaxation session, we gave the same questionnaire to the group as was provided at T0. At T1, the results thus show a different distribution of group members. The first group, which constituted 27% of the students at T0, represented only 15% at T1. The second group increased from 36% to 47%, while the third group remained at 36–37%. The non-parametric statistical study, using the Wilcoxon matched pairs test confirms a significant difference ($p < 0.01$) between the T0 and T1 tests for the group of 22 students. Regarding the comparison of the three groups formed at T0 according to the scale of Cohen et al., despite specific trends, significant differences are revealed at $p < 0.05$ only for groups 2 and 3.

**Table 7.** Stress level before and after a relaxation session.

| Stress Indicators (Cohen et al.) | T0 | | T1 | Wilcoxon Matched Pairs Test |
|---|---|---|---|---|
| More than 27 points | 27.3% | | 15.3% | $p < 0.05$ |
| Between 21 and 26 points | 36.3% | | 47.4% | $p < 0.05$ |
| Less than 21 points | 36.4% | | 37.3% | (NS) |

This survey conducted after the first relaxation session shows a significant decrease in the level of perceived stress among participating students. Confirming this, responses to the questions concerning their general feeling reveal that almost 85% of the students who took part in the session believe that relaxation can sustainably lower their stress levels.

Of course, a single relaxation session is not enough to confirm a downward trend in the degree of perceived stress among students. Further sessions and stress measurements should be carried out over the coming months and years to see if the reduction in perceived stress levels is real and long-lasting.

## 6. Conclusions

Student stress is a growing concern for the public sector and healthcare organisations. Aggravated by the pandemic, stressful situations have an even greater impact on the already vulnerable and exposed student population (studies have shown that while young people are affected by stress, students are even more affected). Our study, using a qualitative approach, sought to understand the causes and symptoms of stress in a selected population in northern France. As such, its results are consistent with other studies in the field. Beyond these initial observations, our intention in this article was to highlight the dialectic link between feelings of stress and relaxation practices. Using a quantitative questionnaire method, we interviewed sports science students who both practised and did not practise relaxation. This allowed us to highlight the strong correlation between these two factors. Although the practice of relaxation is still not very widespread in the university world, our survey shows that there is nevertheless a real interest in further developing the practice by incorporating it into the degree syllabus and general course organisation. Of course, we are aware that our study has one major limitation. Despite the organisation of an immersive session which confirms the positive effects of relaxation, our study does not allow us to demonstrate the lasting effects of relaxation over the medium and long term. This is why a series of relaxation sessions have been scheduled at a frequency of one session per month

for a total of two years (the training lasts three years, so it is possible to keep the same group). After each session, an evaluation of the level of stress will be carried out and will make it possible to assess the lasting effects of relaxation on stress reduction. This study calls for a second, more immersive, in situ study, the results of which will be analysed and presented next year.

**Author Contributions:** Supervision, D.B.-M.; methodology and software, D.G.; Validation, P.C. All authors have read and agreed to the published version of the manuscript.

**Funding:** This work was supported by the URePSSS laboratory and the SHERPAS team and Funding for Open Access Charge by the University of Lille.

**Institutional Review Board Statement:** Not applicable.

**Informed Consent Statement:** Not applicable.

**Data Availability Statement:** Not applicable.

**Conflicts of Interest:** The authors declare no conflict of interest.

## Appendix A

A reference model for qualitative and quantitative analyses based on the Perceived Stress Scale (PSS) by Cohen et al. (1983).

---

Tell us about yourself

**1. How old are you?**
**2. What year of study are you in?**
**3. What is your degree in?**

Your state of stress

**4. In the last month, how many times were you troubled by an unexpected event?**
1 Never
2 Almost never
3 Sometimes
4 Somewhat often
5 Often

**5. In the last month, how many times did you find it hard to control the most important aspects of your life?**
1 Never
2 Almost never
3 Sometimes
4 Somewhat often
5 Often

**6. In the last month, how many times did you feel nervous or stressed?**
1 Never
2 Almost never
3 Sometimes
4 Somewhat often
5 Often

**7. In the last month, how many times did you feel confident enough to deal with any personal issues you may have had?**
5 Never
4 Almost never
3 Sometimes
2 Somewhat often
1 Often

---

**8. In the last month, how many times did you feel like things were going according to plan?**
5 Never
4 Almost never
3 Sometimes
2 Somewhat often
1 Often

**9. In the last month, how many times did you feel incapable of doing everything you needed to do?**
1 Never
2 Almost never
3 Sometimes
4 Somewhat often
5 Often

Your experience of stress

**10. What are the factors or aspects of university life which cause you stress most often? Rank the stressors below from highest to lowest.**

Exams, administrative organisation, personal organisation, workload, uncertainty about the future, change of institution

Your means against stress

**11. What do you do to de-stress?**
See friends, music, sport, walking, going out, sleeping.

Sport to counteract stress

**12. Do you think that sport or exercise is an effective way to combat stress? If you agree, tell us why: If you disagree, tell us why: What sport(s) do you practise?**

Relaxation to counteract stress

**13. What do you know about relaxation?**
(Examples: techniques such as yoga, meditation, mindfulness, etc.).

**14. Have you ever practised relaxation?**

**15. If you have, tell us where and how?**

**Notes**

1   https://www.laboratoire-lescuyer.com/blog/nos-conseils-sante/5-chiffres-surprenants-sur-le-stress (accessed on 12 September 2022).

2   https://www.enseignementsup-recherche.gouv.fr/fr/les-etudiants-inscrits-dans-les-universites-francaises-en-2020-2021-8234 2 (accessed on 27 November 2022).

3   https://www.enseignementsup-recherche.gouv.fr/fr/les-effectifs-d-etudiants-dans-le-superieur-en-2018-2019-en-progression-constante-47408 (accessed on 20 November 2022).

4   France is following the objectives set by the European Union to reach a target of 45% of young adults aged 25–34 with a higher education qualification by 2030 (https://publication.enseignementsup-recherche.gouv.fr/eesr/FR/T666/le_niveau_d_etudes-_de_la_Population_et_des_jeunes/ (accessed on 19 October 2022).

5   Macotta and Corvos (2017).

6   Student health in 2007. Fifth survey of the national union of regional student mutual insurance companies, (June 2007). Study conducted on 14,000 student volunteers (Île-de-France).

7   OVE (National Observatory on Student Life) survey on the living conditions of students in 2020 (Panel of over 100,000 students who responded to the survey)—http://www.ove-national.education.fr (accessed on 5 February 2023).

8   L'état de l'Enseignement supérieur et de la Recherche en France, n°8, juin 2015. https://publication.enseignementsup-recherche.gouv.fr/eesr/8/EESR8_ES_14-la_vie_etudiante_la_sante_des_etudiants.php (accessed on 5 September 2022).

9   Enquête pour le Baromètre l'Étudiant-BVA: les jeunes stressés mais optimistes (Baromètre l'Étudiant-BVA Survey: Young People Stressed Yet Optimisitc): https://www.letudiant.fr/lifestyle/Sante-mutuelle-et-assurance/barometre-l-etudiant-bva-les-jeunes-stresses-mais-optimistes.html (accessed on 11 September 2022).

10   Goulard, F., 2007. "L'Enseignement supérieur en France, état des lieux et propositions" ("Higher Education in France: state of play and proposals"). Report by the Minister for Education and Research.

11   Survey for the website l'Étudiant conducted by A. Petitdemange in November 2020. «Crise sanitaire: la précarité des étudiants augmente» ("Health crisis: student insecurity on the rise"): https://www.letudiant.fr/lifestyle/aides-financieres/crise-sanitaire-la-precarite-des-etudiants-augmente.html (accessed on 12 October 2021).

12   FAGES-IPSOS survey, «L'impact de la crise sur la santé des jeunes» ("The impact of the crisis on young people's health"). Study conducted in June 2020. Press release: https://www.fage.org (accessed on 8 March 2022).

13   2020 study of 70,000 students by the French National Centre for Resources and Resilience and the FHF Research & Innovation Fund: https://www.reseau-chu.org/article/enquete-le-lourd-impact-du-confinement-sur-la-sante-mentale-des-etudiants0/ (accessed on 18 February 2023).

14   INSEE survey, 2016, "Revenu, niveau de vie et pauvreté en 2016" ("Income, living standards and poverty in 2016"). https://www.insee.fr/fr/statistiques/3650242?sommaire=3650460#titre-bloc-3 (accessed on 8 January 2021).

15   Ipsos survey report, 2020, "Impact de la crise du COVID-19 sur les étudiants" ("Impact of the COVID-19 pandemic on students"). https://www.fage.org/ressources/documents/3/6079-Note-FAGE-Impact-de-la-crise-sur-le.pdf (accessed on 30 March 2020).

16   MILDECA survey report, 2021, "COVID-19: isolement et conditions de travail favorisent les conduites addictives" ("COVID-19: isolation and working conditions favour addictive behaviour"). https://www.drogues.gouv.fr/covid-19-isolement-et-conditions-de-travail-favorisent-les-conduites-addictives (accessed on 6 May 2020).

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
