# Peer review of "Student Stress and the Effects of Relaxation: A Study Conducted at the University of Lille in Northern France"

_socsci, doi:10.3390/socsci12060318_

Round 1
Reviewer 1 Report
The study is interesting, but given the amount of work involved, it is a pity to analyse stress only in terms of the type of study.
Author Response
We thank the reviewers for their analysis. The critics of the reviewers helped us to improve our paper. Several points have been highlighted and we have tried to respond to each criticism and recommendation:
About the general structure of the paper
- We have overall improved the general format of the paper.
- The citations have been revised to be more appropriate to the content.
- The sources of the tables have been clarified. We have given a proper explanation of the figures mentioned in the paper with the sources.
- The statistical results have been removed from the discussion section and added to the results section.
- We have corrected typos and grammatical errors.
About abstract
- The abstract was not informative enough. We have modified and completed it, adding details about the sample, the method used and the main results obtained.
- We have better presented the tables and figures. We have translated the words in French.
Introduction
- We have better contextualised previous studies and theoretical researches.
- We have improved the link between the article and previous studies.
About results section
- For the Empirical Research, the results are more clearly presented with an effort of clarification.
About methodology
- We have better defined the study population and clarified the choice of sample.
- We have clarified the verification tests to assess the reliability of the results obtained, in particular on the question of significant differences between relaxation practitioners and non-practitioners on their stress level,
In conclusion, we have further clarified the limitations of our study and explained the future fields of application for future studies.
The authors
Reviewer 2 Report
A paper's formatting needs improvement.
The citations must be written appropriately.
Correct grammatical and typing mistakes.
Write theoretical background and mention previous studies on 'Student stress' and 'relaxation'. Moreover, try to link the cited studies.
Define the population of the study.
What is the sample selection criteria?
Write sources of tables.
To report a significant difference between practitioners and non-practitioners of relaxation in terms of their stress levels which statistical test was used in the study?
Write and explain a proper description of the figure mentioned in the paper with the source.
Mention limitations and future scope of study with practical applicability.
Author Response

(The authors gave the same response as above.)

Reviewer 3 Report
Dear Authors,
Please find my comments in the file attached

Author Response

(The authors gave the same response as above.)

Round 2
Reviewer 3 Report
Dear Authors,
There is no revised version of the manuscript so I can't check which alterations were made in the body of the manuscript. Instead of the revised version of the manuscript there is a paper confirming language proofreading.
Author Response
We are sorry. There was a problem loading the file. I hope this error is now fixed.
We thank the reviewer for his analysis. The critics of the reviewer helped us to improve our paper. Several points have been highlighted and we have tried to respond to each criticism and recommendation:
About the general structure of the paper
- We have overall improved the general format of the paper.
- The citations have been revised to be more appropriate to the content.
- The sources of the tables have been clarified. We have given a proper explanation of the figures mentioned in the paper with the sources.
- The statistical results have been removed from the discussion section and added to the results section.
- We have corrected typos and grammatical errors.
About abstract
- The abstract was not informative enough. We have modified and completed it, adding details about the sample, the method used and the main results obtained.
- We have better presented the tables and figures. We have translated the words in French.
Introduction
- We have better contextualised previous studies and theoretical researches.
- We have improved the link between the article and previous studies.
About results section
- For the Empirical Research, the results are more clearly presented with an effort of clarification.
About methodology
- We have better defined the study population and clarified the choice of sample.
- We have clarified the verification tests to assess the reliability of the results obtained, in particular on the question of significant differences between relaxation practitioners and non-practitioners on their stress level,
In conclusion, we have further clarified the limitations of our study and explained the future fields of application for future studies.
Round 3
Reviewer 3 Report
Dear Authors,
Find the answers and comments in the file attached.

Author Response
We thank you for your comments and criticisms concerning the improvement of the article. Thanks again for your review and advice.
Initially, we took over all the tables to enrich and improve them. Indeed, terms in French were still present in some tables. We deleted a table that was more like a figure 1 [page 10] than we realized.
As you have noted, the references to the tables in the text were not appropriate. We have corrected these errors.
Rightly, the statistical analyses were not rigorous enough and sometimes did not match what was explained in the text.
In a second step, ultimately very substantial, we took over all the initial matrices to redo and better adapt the statistical analyses, mainly of the non-parametric type as explained, the distributions of the variables sometimes not following normal laws.
The authors have improved the manuscript however the major issues remain because inadequate approaches to the analysis of empirical data are applied.
Comments and suggestions:
- Table 2. As far as I understand, the number of stress factors (SF or FS?) was recorded for each participant. This is a quantitative variable. And it is necessary to compare samples for this quantitative variable using parametric or nonparametric/rank comparison methods to determine the statistical significance of the differences.
Response: In relation to this remark associated with Table 2, we performed the non-parametric Kruskal-Wallis Anova test.: “ After noting that the distributions of the variables did not follow a normal distribution (Shapiro-Wilk test), the application of the non-parametric Kruskal-Wallis ANOVA test by rank reveals strong significant differences at p<0.001 between the disciplinary groups. Multiple comparisons (p values and z' values) of the sum of stress factors characterize and confirm (p<0.001) significant differences mainly between the Medicine and Law groups with the other groups, STAPS, engineering and business schools.”
- Table 3. There is no reference to Table 3 in the text.
Response: We corrected and changed the indexing of the tables and corrected/added the indexing in the text.
For each participant the following variables were recorded: Group, Exam (0 – no, 1 – yes), etc. Then to estimate the difference between the groups (stat. confidence) by variable, for example, Exam, you need to provide us with the contingency table Group * Exam, etc.
The same remarks apply to the tables 4, 5…
Response:We have corrected and improved tables 4, 5, 6.
We thought of making correlational matrices allowing to weight all the variables, according to the different groups (Sports Science, Medicine, ...) according to the variable perceived stress state , but this could generate cumbersomeness in the constitution and perception of the tables as well as in the resulting interpretations and additional analyzes.
- «When asked the question “in the last six months, how often did you feel stressed: never, sometimes, often or very often?”… To answer this question, we conducted a cross-tabulation of stress levels between relaxation practitioners and non-practitioners. At p<0.001 of t-test of Student, our analysis confirms that there is a significant difference between practitioners and non-practitioners of relaxation in terms of their stress levels». This is not a cross tabulation, but a comparison of 2 samples by stress level. Were the averages compared (once the t-test of Student is mentioned)? Or is it still a contingency table? Then what does t-test have to do with it?
Response: We have completely redone Table 6 to enrich it with an adapted statistical analysis (Kruskal-Wallis ANOVA by Ranks and Chi-Square tests between the "stress" factor and the "practices relaxation" factor). The statistical analysis shows significant differences between students practicing and non-practicing relaxation and stress.
The text corresponding to this interpretation has been corrected and improved: “When it comes to relaxation techniques, although more than half of students had never practised these methods (58%, of which one third are girls and two thirds are boys), does our study reveal relaxation to have a beneficial effect on reducing stress among students? To answer this question, we performed a Kuskal-Wallis ANOVA test by rank associated with a Cki-Square test to compare stress levels between practitioners and non-practitioners of relaxation. Our analysis confirms that there is a significant difference between practitioners and non-practitioners of relaxation in terms of their stress levels (p<0,01). Indeed, among the 41% of students who report never having felt stressed, 52% practise relaxation (of which 43% are girls). However, of the 17% who say they are very often stressed, only 11% practise relaxation (of which 15% are girls). As can be seen, the group reporting never having felt stressed had the highest proportion of students practising relaxation techniques (52%). On the other hand, the majority of students who report often and very frequently stressed (59%) very often do not practise relaxation (67%).”
- Table 6. What does correlation have to do with it? Between which TWO variables was the correlation calculated? The contingency table 2x2 is shown. It should contain not only percentages, but also frequencies. And the corresponding method is applied for conjugacy tables.
Response: As said in the previous paragraph, we have reworked table 6.
We have enriched Table 7 with a before-and-after comparison to provide statistical justification for significant differences, as well as the corresponding text: " The non-parametric statistical study, using the Wilcoxon matched pairs test confirms a significant difference (p<0.01) between the T0 and T1 tests for the group of 22 students. Regarding the comparison of the 3 groups formed at T0 according to the scale of Cohen et al., despite specific trends, significant differences are revealed at p<0.05 only for groups 2 and 3.”
Looking forward to your comments, please accept our best regards.